# Pressure- and 3D-Derived Coronary Flow Reserve with Hydrostatic Pressure Correction: Comparison with Intracoronary Doppler Measurements

**DOI:** 10.3390/jpm12050780

**Published:** 2022-05-12

**Authors:** Balázs Tar, András Ágoston, Áron Üveges, Gábor Tamás Szabó, Tibor Szűk, András Komócsi, Dániel Czuriga, Benjamin Csippa, György Paál, Zsolt Kőszegi

**Affiliations:** 1Kálmán Laki Doctoral School of Biomedical and Clinical Sciences, University of Debrecen, 4032 Debrecen, Hungary; b.tar@upcmail.hu (B.T.); agostonandras48@gmail.com (A.Á.); aronaok@gmail.com (Á.Ü.); nszgt@med.unideb.hu (G.T.S.); tszuk01@gmail.com (T.S.); dczuriga@med.unideb.hu (D.C.); 2Szabolcs–Szatmár–Bereg County Hospitals, University Teaching Hospital, 4400 Nyíregyháza, Hungary; 3Institute of Cardiology, University of Debrecen, 4032 Debrecen, Hungary; 4Heart Institute, Medical School, 7624 Pécs, Hungary; komocsi@gmail.com; 5Department of Hydrodynamic Systems, Budapest University of Technology and Economics, 1111 Budapest, Hungary; bcsippa@hds.bme.hu (B.C.); paal@hds.bme.hu (G.P.)

**Keywords:** stable angina, fractional flow reserve (FFR), coronary flow reserve (CFR), quantitative coronary angiography, coronary microvascular disease, microvascular resistance reserve (MRR)

## Abstract

Purpose: To develop a method of coronary flow reserve (CFR) calculation derived from three-dimensional (3D) coronary angiographic parameters and intracoronary pressure data during fractional flow reserve (FFR) measurement. Methods: Altogether 19 coronary arteries of 16 native and 3 stented vessels were reconstructed in 3D. The measured distal intracoronary pressures were corrected to the hydrostatic pressure based on the height differences between the levels of the vessel orifice and the sensor position. Classical fluid dynamic equations were applied to calculate the flow during the resting state and vasodilatation based on morphological data and intracoronary pressure values. 3D-derived coronary flow reserve (CFR_p-3D_) was defined as the ratio between the calculated hyperemic and the resting flow and was compared to the CFR values simultaneously measured by the Doppler sensor (CFR_Doppler_). Results: Haemodynamic calculations using the distal coronary pressures corrected for hydrostatic pressures showed a strong correlation between the individual CFRp-3D values and the CFR_Doppler_ measurements (*r* = 0.89, *p* < 0.0001). Hydrostatic pressure correction increased the specificity of the method from 46.1% to 92.3% for predicting an abnormal CFR_Doppler_ < 2. Conclusions: CFR_p-3D_ calculation with hydrostatic pressure correction during FFR measurement facilitates a comprehensive hemodynamic assessment, supporting the complex evaluation of macro-and microvascular coronary artery disease.

## 1. Introduction

According to the current European guideline on coronary revascularization, pressure wire-derived fractional flow reserve (FFR) measurement is recommended for the functional assessment of lesion severity in patients with 40–90% diameter stenosis and without prior evidence of ischemia [1]. A more recent guideline suggests the consideration of a guidewire-based coronary flow reserve CFR measurement in patients with persistent symptoms but with preserved FFR [2] based on earlier publications [3,4,5]. The combination of FFR and CFR evaluation may identify the potential components of ischemia originating from the decreased conductance of the epicardial vessels and the increased resistance of the microvasculature [6,7,8,9].

As a temperature sensor, the pressure-wire sensor makes it possible to calculate thermodilution; however, the bolus method comes with several limitations, as already detailed in early validation studies [10,11,12]. On the other hand, the direct measurement of coronary flow velocity by a Doppler sensor is considered technically difficult to perform; consequently, it is not routinely used in clinical practice.

The resistance of the microvasculature (MR) is defined as the ratio of the distal coronary pressure divided by the distal coronary flow rate. The resistive reserve ratio (RRR) is the index expressing the ratio between basal and hyperemic microcirculation resistance (bMR divided by hMR) [13,14].

Lately, the term microvascular resistance reserve (MRR) was suggested for specific characterization of the microvasculature [15]:MRR = (CFR/FFR) × (Pa rest/Pa hyp)(1)
where Pa rest and Pa hyp are the aortic pressures during resting and hyperemic state.

The concept was proposed in connection with the continuous thermodilution technique (requiring a special infusion catheter) and was validated by intracoronary Doppler measurements [16].

In our study, we aimed at developing a clinically applicable method for calculating specific CFR and RRR values (CFR_p-3D_ and RRR_p-3D_) during FFR measurement, using simple hemodynamic calculations that combine intracoronary pressure data and 3D anatomical parameters (Figure 1). The results of our calculations were compared to data obtained using invasive Doppler wire measurement, as a gold standard of flow assessment.

It has recently been underlined that pressure differences are systematically detectable between the different segments of the coronary arteries in the supine position [17,18,19]. We also investigated how the correction of distal pressure for hydrostatic pressure offset affects the pressure-derived flow determinations.

## 2. Materials and Methods

### 2.1. Patient Inclusion and Exclusion Criteria

Patients, who underwent clinically indicated invasive physiological investigations, were selected for this study, with single stenosis of intermediate severity (40–80% based on a visual assessment) in a main branch of the epicardial coronary artery system. Cases with good quality hyperemic and resting pressure and Doppler traces were included for the evaluations. Only traces without pressure drift (<1 mmHg) confirmed by the pullback of the pressure sensor at the end of the procedure were considered. Patients with an acute coronary syndrome, left main stenosis, ostial stenosis, earlier bypass surgery, or diffuse coronary artery disease were excluded. The study has been approved by the local ethics committee of the University of Debrecen and has therefore been performed in concordance with the Declaration of Helsinki.

### 2.2. Invasive Coronary Angiography and Simultaneous Pressure and Flow Measurement by ComboWire

After administering 5000 international units (IU) of intravenous, unfractionated heparin (UFH) and intracoronary glyceryl trinitrate (GTN), diagnostic angiographic cine-recordings were acquired from standard projections, using digital X-ray equipment (Axiom Artis, Siemens, Erlangen, Germany). Diagnostic angiographic images were recorded at 15 frames per second. Low- or iso-osmolar contrast material (CM) (iopamidol (Scanlux) or iodixanol (Visipaque)) was injected in 5 mL fractions with a speed of 3 mL/sec using a dedicated contrast pump (ACIST CVi™, ACIST Medical Systems, Eden Prairie, MN, USA). If the operator detected a 40–80% diameter stenosis by visual assessment, complete physiological measurements were performed via a 6F guiding catheter, using a ComboWire equipped with both pressure and Doppler sensors (Philips Volcano, San Diego, CA, USA).

After the pressures were equalized with the sensor positioned at the level of the catheter tip, it was advanced through the coronary artery stenosis, and measurements were performed approximately 2 cm distal to the lesion. Following the basal pressure and flow measurements, 150–200 µg intracoronary adenosine was administered, and the pressure and Doppler traces were recorded. One representative measurement is presented in Figure 2.

### 2.3. Three-Dimensional Quantitative Coronary Artery Reconstruction and Hemodynamic Calculations

Offline 3D angiographic reconstruction was performed from two selected angiograms of good quality, with at least 25° difference in angle, using dedicated software (QAngio XA^®^ Research Edition 1.0, Medis Specials bv, Leiden, The Netherlands). The reconstructed vessel segment was marked from the coronary orifice to the location of the wire sensors. Numerous geometric measures describing the lesion (average cross-sectional diameters and vessel segment lengths), as well as the proximally and distally connecting vessel segments, were automatically obtained by the software. These values with intracoronary pressure at the proximal and distal positions during the resting and vasodilation states were combined for hemodynamic calculations. The method and its validation are described in our previous papers in detail [20,21], and an online calculation tool (http://coronart.unideb.hu/) (accessed on 5 March 2022) is available.

### 2.4. Calculation of the Doppler-Derived Indices

The resistance of the microvasculature (MR) was defined as the ratio of the distal coronary pressure divided by the distal coronary flow rate both in the basal state and during hyperemia:(2)$$bMR=\frac{\mathrm{Pd}\;\mathrm{rest}}{APV\;B}$$
(3)$$hMR=\frac{\mathrm{Pd}\;\mathrm{hyperemic}}{APV\;P}$$
where *bMR*: basal microvascular resistance, *hMR*: hyperemic microvascular resistance and *APV-B*: basal average peak velocity, *APV-P*: peak average velocity measured by the ComboWire during basal and hyperemic flow (see Figure 2).

The resistive reserve ratio (RRR) as the index of the ratio between basal and hyperemic microcirculation resistance [13,14] was calculated as follows:(4)$$\mathrm{RRR}\;\mathrm{Doppler}\;=\frac{bMR}{hMR}$$

### 2.5. Calculation of the RRR_p-3D_

The RRR was also defined analogously from the calculated flows defined by simple flow equations using the pressure and 3D anatomy data (*Q_p-3D_*):(5)$$\mathrm{RRR}\;p3D=\frac{\mathrm{Pd}\;\mathrm{rest}\;/\;Qp3D\;rest}{\mathrm{Pd}\;\mathrm{hyperemic}\;/\;Qp3D\;hyperemic}$$

### 2.6. Correction of the Distal Coronary Pressure for Hydrostatic Pressure

In the supine position, the measured pressure difference between the catheter tip and the pressure sensor distal to the lesion originates from two components, namely the pressure loss caused by the flow through the stenosis, the difference between the hydrostatic pressure at the catheter tip at the coronary orifice, and the level of the distal intracoronary sensor (Figure 3).

The latter component can be referred to as hydrostatic offset (∆P hydrostatic pressure) and can modify the detected pressure ratio values through the “altered” distal pressure value [18,19].

The correction of distal pressure for hydrostatic pressure (Pd corr) was based on the height differences between the orifice and other coronary artery segments in supine positions. The distal pressure values were corrected, using a correction factor of 0.77 mmHg hydrostatic pressure per 1 cm height difference, where blood density was taken as 1050 kg/m^3^ (Figure 3):Pd corr = Pd − ∆P hydrostatic pressure(6)

### 2.7. Statistical Analysis

Statistical evaluations were performed in MedCalc Statistical Software, Version 14.8.1 (MedCalc Software bvba, Ostend, Belgium). Following a normality test, Spearman’s correlation analysis was carried out. The correlation between CFR_p-3D_ and the CFR_Doppler_ was examined both without and with hydrostatic pressure correction of the distal pressure. The agreement between CFR_Doppler_ and CFR_p-3D_ was assessed using the Bland-Altman analysis. The area under the curve (AUC) calculated by receiver operating characteristic analysis was applied to determine the diagnostic power of CFR_p-3D_ without and with hydrostatic pressure correction. The sensitivity and specificity of CFR_p-3D_ without and with hydrostatic pressure correction were calculated using the standard method.

## 3. Results

We performed simultaneous intracoronary pressure and Doppler measurement by ComboWire in 20 patients screened in the study. In 3 cases the Doppler signal quality was insufficient for the calculation, in 1 further case more than 2 mmHg drift was detected at the end of the investigation and the attempt for repeat measurement also failed. Therefore, sixteen 16 patients (14 males, 2 females) with single, intermediate epicardial coronary stenosis were involved in the study. In 3 cases, measurements were performed both before and after stent implantation. Patient characteristics are presented in Table 1. The results of 3D reconstruction and the measured physiological data are summarized for each interrogated vessel in Table 2.

### 3.1. Correlation and Agreement between the Results of the CFR_Doppler_ Measurements and Calculated CFR_p-3D_ Values without and with the Correction for Hydrostatic Offset

When including morphological data from 3D coronary angiography in the hemodynamic calculation and correcting the values for hydrostatic pressure, a strong correlation was found between the individual CFR_p-3D_ values and the CFR_Doppler_ measurements (*r* = 0.89, *p* < 0.0001). A weak but still significant correlation was demonstrated even without the correction of hydrostatic error (*r* = 0.57, *p* = 0.01) (Figure 4A,B). The difference between the two correlations was found to be significant (*p* = 0.02).

The Bland-Altman analysis showed the mean differences between the Doppler-measured and the calculated CFR_p-3D_ values with and without hydrostatic offset correction to be −0.02 (±1.96 SD: 0.47, −0.50) and −0.05 (±1.96 SD: 1.38, −1.48), respectively. After hydrostatic offset correction, the values of CFR_p-3D_ and those of CFR_Doppler_ got closer without any systematic skewing suggesting a higher level of concordance (Figure 4C,D).

### 3.2. Correlation and Agreement between the Results of the ComboWire Based RRR Measurements (RRR_Doppler_) and the Calculated RRR_p-3D_ Values with the Correction for Hydrostatic Offset

The calculated microvascular resistance reserve (RRR_p-3D_) also demonstrated a good correlation with the measured RRR_Doppler_ values (*r* = 0.83, *p* < 0.0001) Figure 5A. The Bland-Altman analysis showed the mean differences between the Doppler-measured and the calculated RRR_p-3D_ values with hydrostatic offset correction to be −0.03 (±1.96 SD: 0.63, −0.68) Figure 5B.

### 3.3. The Results of Hydrostatic Offset Correction on the Pressure Ratios and the CFR_p-3D_ in the Main Coronary Branches

Figure 6 shows the clustered multiple variable graphs of resting Pd/Pa (A), FFR (B), and the CFR_p-3D_ (C) without and with hydrostatic pressure correction. In line with the findings of our previous work [18], the correction of the hydrostatic offset resulted in specific concordant differences between the uncorrected and corrected values in the main coronary branches in both resting and hyperemic (FFR) states (Figure 6A,B). The correction definitively increased the values in the LAD, while in the LCx and the RCA, the values decreased. We observed much higher differences in CFRs, especially in the range of higher CFR values (Figure 6C).

### 3.4. Diagnostic Powers of CFR_p-3D_ Calculated from the Distal Pressure without and with Hydrostatic Offset Correction for Identifying CFR_Doppler_ < 2

The diagnostic power of different computations of the CFR_p-3D_ for predicting the abnormal CFR_Doppler_ was assessed using the computed CFR_p-3D_ (cut-off value = 2). The AUCs of the values calculated without and with hydrostatic error correction was 0.73 (CI: 0.48–0.90) and 0.96 (CI: 0.78–1.00), respectively. Correcting for hydrostatic pressure offset increased the specificity of the method from 46.1% to 92.3%, while the sensitivity of both calculations was 100%.

## 4. Discussion

In pioneering research, the pressure drop across arterial stenosis was estimated satisfactorily by simple flow equations [22]. Later the 3D anatomical characteristics of the coronary artery were also incorporated into computational fluid dynamics calculations leading to the virtual, image-based FFR assessment [23,24,25]. Recently, the possibility to determine coronary flow from invasively measured intracoronary pressure has arisen by “backward” calculations [26,27]. The so-called pressure-bounded coronary flow reserve (CFR_pb_) assessment identified the possible range of CFR according to the resting and hyperemic pressures.

Wijntjens and colleagues compared the CFR_pb_ to flow-derived CFR defined by thermodilution and Doppler measurements in 453 intermediate coronary lesions, but they found a poor diagnostic agreement between the two estimations [28]. It is important to emphasize, that in this publication, hydrostatic offset correction of the distal coronary pressure was not applied to CFR calculations.

Similar to the method presented in this article, an absolute flow calculation with fluid dynamic computation (CFD) using the Ansys software (QCFD) was recently published by Morris et al. [29]. In contrast with our method where the distal flow is rendered to the tapered vessel size [20], their in vitro and in vivo models did not account for flow to side branches, resulting in underestimation of the volumetric flow [30]. This underestimation could lead to unlikely low resting and hyperemic calculated flow values in major coronary branches, as was pointed out in the editorial responding to their paper [31]. It is very obvious that in their in vivo study, the hydrostatic pressure error had caused, at least partly, a very weak correlation to the Doppler results.

The direction of the effect of the hydrostatic offset depends on the orientation of the sensor in the distal position relative to the coronary orifice.

If one interrogates distal LAD with the sensor, the hydrostatic pressure is lower in the supine position, which results in higher pressure ratios after hydrostatic offset correction. In contrast, LCx takes a downward course, which leads to higher hydrostatic pressure at the level of the sensor, and consequently, the pressure values are lower compared to the one measured following correction. The height correction of RCA measurements can result in a slight increase of the distal pressure value, as the distal sensor in the distal RCA is at a lower level compared to the orifice (Figure 6A,B) [18]. Thus, a slight increase in the corrected pressure ratios can be observed (Figure 6A,B).

In our opinion, the correction of distal pressure for hydrostatic pressure is essential when determining pressure-derived CFR. A minor hydrostatic pressure may have a significant influence on the measured pressure gradient, especially in the resting state.

This phenomenon is demonstrated in Figure 6C, where the correction resulted in significant differences between the calculated CFR_p-3D_ and the uncorrected values, most prominently in the range of higher CFR values.

The CFR_p-3D_ values calculated after the correction for hydrostatic pressure and those derived from native pressure values were compared with the Doppler flow measurements. A strong correlation was demonstrated between the individual CFR_p-3D_ and the CFR_Doppler_ values when the correction for hydrostatic pressure was made, while the only weak correlation was found without hydrostatic pressure correction.

Importantly, the elimination of hydrostatic pressure offset increased the specificity of our method from 46.1% to 92.3%, while the sensitivity of both calculations remained 100% against the “gold standard” Doppler measurement.

## 5. Limitations of the Study

The main limitation of our pilot study of CFR_p-3D_ calculations is represented by the small sample size; however, the archived and statistically highly significant results look promising.

We are aware that our simple model considers only Hagen-Poiseuille-type friction losses and highly simplified Borda-Carnot-type separation losses. For this reason, the calculation of the flow rate is also not expected to be always accurate, but because the CFR is by definition a ratio-type parameter, the CFR_p-3D_ may be accurate enough for clinical applications [22].

The simplified hemodynamic model used for the calculation of the CFR_p-3D_ can consider only one stenosis, with normal proximal and distal segments. Consequently, our flow calculation method in the present form may not be adequate for assessing the hemodynamic relevance of sequential stenoses.

In cases with a very low resting pressure gradient, any small error during the measurement could potentially cause a great deviation in the results, as these values are represented in the denominator during the calculations. However, most of the cases with intermediate coronary lesions showed not less than a 1–2 mmHg resting pressure gradient, which allowed the appropriate calculation of the CFR_p-3D_.

## 6. Conclusions

In this study, we proposed a method of combined determination of FFR and CFR/RRR without the need for a Doppler wire or thermodilution procedure. In our opinion, the CFR_p-3D_ is applicable for most coronary angiography with the clinically indicated invasive measurement of the FFR, when the target vessel is suitable for 3D reconstruction. The flow calculation does not require significantly more time this way. As a result, the consequences of epicardial stenosis can be assessed simultaneously with the state of the microvasculature, thereby supporting the clinical decision for selecting the most appropriate therapy. In our opinion, large-scale studies are warranted to investigate the clinical relevance of the pressure-flow relation determined by our technique [32].

## 7. Patents

The patent of the method detailed in this paper has been issued by the European Patent Office (WO2019175612, applicant: University of Debrecen, inventor: Z.K.).

## Figures and Tables

**Figure 1 jpm-12-00780-f001:**
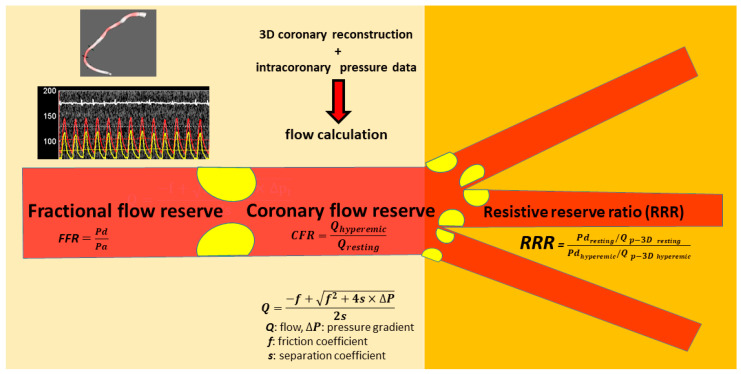
Calculation of CFR and RRR values during FFR measurement. The method for calculating CFR_p-3D_ and RRR_p-3D_ uses hemodynamic calculations combining intracoronary pressure data (top left panel) and 3D anatomical parameters (bottom left panel). Based on the hyperemic and resting pressure data, as well as 3D anatomical parameters, simple hemodynamic equations were used to calculate resting and hyperemic flow, CFR, and RRR. The detailed description of the flow calculations is described in the patent of the method: https://patents.google.com/patent/WO2019175612A2/en (accessed on 5 March 2022).

**Figure 2 jpm-12-00780-f002:**
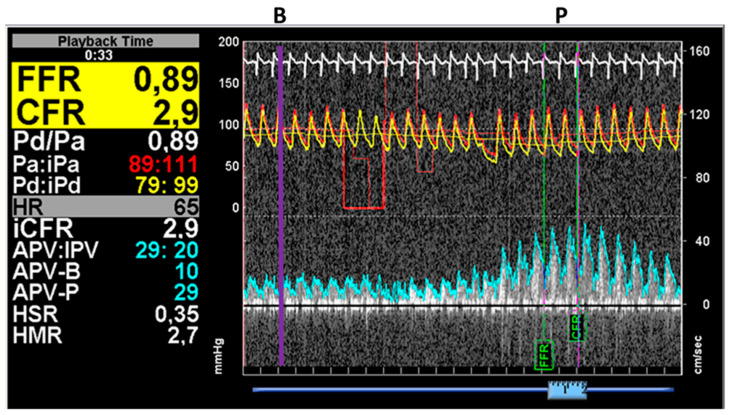
Results of simultaneous pressure and flow measurements by the ComboWire. In this case, the average proximal (aortic) and distal pressures were detected to be 95 mmHg and 88 mmHg, respectively. At maximal hyperemia (P), the average peak velocity (APV-P) increased to 29 cm/s from the basal (B) velocity of 10 cm/s (APV-B) parallel with the increase in the pressure drop (the proximal and distal pressures were 89 mmHg and 79 mmHg, respectively). The measured FFR was 0.89, while the CFR was 2.9 (Case 10).

**Figure 3 jpm-12-00780-f003:**
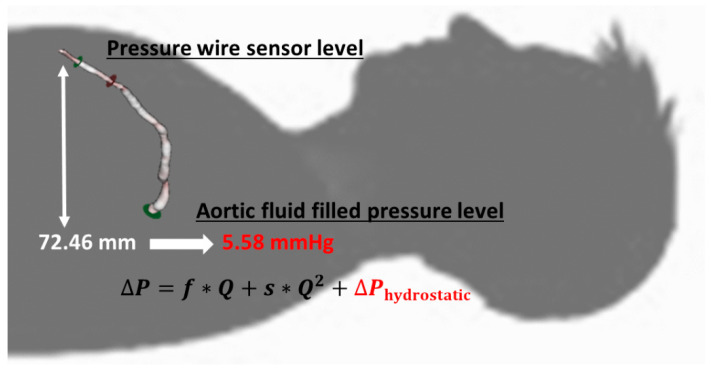
The height difference between the LAD orifice and the sensor position. After 3D reconstruction, the height difference between the orifice and the pressure wire sensor was transformed to mmHg getting hydrostatic pressure (red) (Case 10). This value (5.58 mmHg) influences the gradient between the aortic pressure at the tip of the catheter and the pressure detected by the sensor of the pressure wire, and it has a great impact on the results of the CFR calculation.

**Figure 4 jpm-12-00780-f004:**
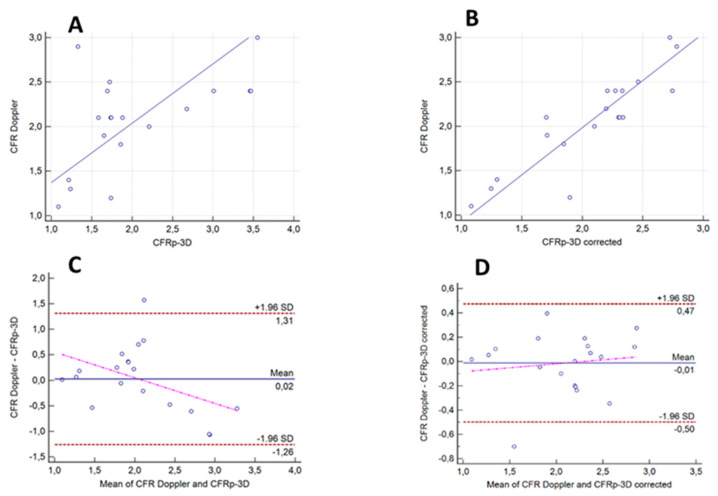
Correlations and agreements between calculated CFR_p-3D_ values without and with hydrostatic offset correction, and the measured Doppler CFR values. (**A**,**B**) Correlations between the calculated CFR_p-3D_ values without and with hydrostatic offset correction and the measured Doppler CFR values: *r* = 0.57, *p* = 0.01, and *r* = 0.89, *p* < 0.0001, respectively. (**C**,**D**) Bland-Altman analysis of the agreement between the calculated CFRp-3D values without and with hydrostatic offset correction and the measured Doppler CFR values.

**Figure 5 jpm-12-00780-f005:**
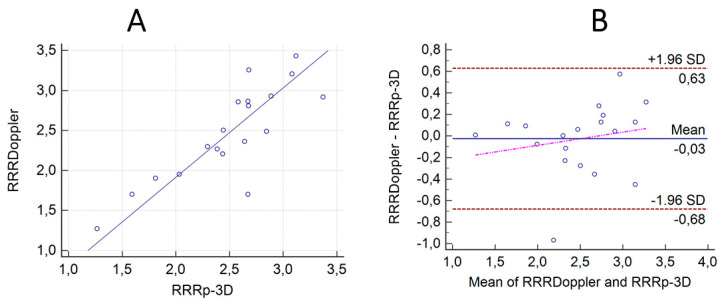
Correlation and agreement between the results of the ComboWire-based RRR measurements (RRR_Doppler_) and the calculated RRR_p-3D_ values with the correction for hydrostatic offset. (**A**) The calculated microvascular resistance reserve RRR_p-3D_ corrected to the hydrostatic pressure error demonstrated a good correlation with the measured MRR_Doppler_ values (*r* = 0.83, *p* < 0.0001). (**B**) The Bland-Altman analysis showed the mean differences between the Doppler-measured and the calculated RRR_p-3D_ values with hydrostatic offset correction to be −0.03 (±1.96 SD: 0.71, −0.78).

**Figure 6 jpm-12-00780-f006:**
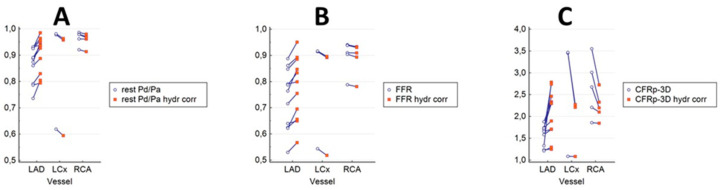
Clustered multiple variable graphs of resting Pd/Pa, FFR, and CFRp-3D without and with hydrostatic pressure correction. In both the resting and the hyperemic (FFR) state (**A**,**B**), the correction of the hydrostatic offset resulted in differences in consequent directions between the uncorrected and corrected values in the main coronary branches. The correction significantly increased the values in the LAD, while in the LC and the RCA the values decreased. Much greater differences in the same direction can be observed for the CFRs, especially in the range of the higher values (**C**).

**Table 1 jpm-12-00780-t001:** Clinical characteristics.

Patient No.	Age	Gender	Target Vessel	Hypertension	DM	Dyslipidaemia	Hyperuricaemia	Chronic Renal Failure	Aorta Stenosis	DVT	CCS (prev)	PCI (prev)	PAD
1	62	f	RCA	√	√	χ	χ	χ	χ	√	√	√	χ
2	51	m	CX	χ	χ	χ	χ	χ	χ	χ	√	√	χ
3	60	m	RCA	√	χ	√	χ	χ	√	χ	√	χ	√
4	66	m	LAD	√	χ	√	χ	χ	χ	χ	√	√	χ
5	65	m	LAD	χ	√	√	χ	χ	χ	χ	√	√	χ
6	55	m	LAD	√	χ	√	χ	χ	χ	χ	√	χ	χ
7	64	m	RCA	√	χ	χ	χ	√	χ	χ	√	√	χ
8	55	m	LAD	√	√	√	χ	√	χ	χ	√	χ	χ
9	69	m	LAD	√	√	χ	χ	χ	χ	χ	√	√	χ
10	43	m	RCA	√	χ	√	√	χ	χ	χ	√	√	χ
11	56	m	LAD	√	√	χ	χ	χ	χ	χ	√	√	χ
12	52	m	LAD	χ	χ	√	χ	χ	χ	χ	√	√	χ
13	66	m	CX-OM	√	√	√	χ	χ	χ	χ	√	√	√
14	60	f	CX-OM	√	√	χ	χ	χ	χ	χ	√	χ	χ
15	63	m	LAD	√	χ	√	χ	χ	χ	χ	√	χ	χ
16	66	m	LAD	√	√	χ	√	χ	√	χ	√	χ	χ

RCA: right coronary artery; CX: circumflex artery; LAD: left descending coronary artery; CX-OM: obtuse marginal branch of circumflex artery; DM: diabetes mellitus; DVT: deep vein thrombosis; CCS: chronic coronary syndrome; PCI: percutaneous coronary intervention; PAD: peripheral artery disease.

**Table 2 jpm-12-00780-t002:** Measured and calculated hemodynamic parameters of the interrogated lesions.

Case No.	Vessel Segment	Hydrostatic Pressure Difference (mmHg) **	Pd/Pa Rest	FFR	CFR_p-3D_	CFR_p-3D_ Corrected ***	APV-B (cm/s)	APV-P (cm/s)	CFR Doppler	RRR_p-3D_ Corrected ***	RRR_Doppler_ Corrected ***
1	RCA med	0.46	0.99	0.94	3.55	2.72	14	42	3	3.12	3.43
2	LCx dist	1.99	0.98	0.92	3.47	2.21	19	45	2.4	2.67	2.86
3	RCA med	0.05	0.96	0.91	1.86	1.85	18	32	1.8	2.03	1.95
4	LAD prox	−2.93	0.92	0.86	1.74	2.3	24	50	2.1	2.44	2.21
5	LAD med	−2.5	0.86	0.63	1.58	1.7	14	29	2.1	2.68	3.26
6	LAD prox	−0.69	0.79	0.64	1.24	1.25	32	42	1.3	1.81	1.91
7 *	RCA med	0.76	0.92	0.79	2.21	2.1	15	30	2	2.39	2.27
8	RCA med (post stent)	0.77	0.98	0.94	2.68	2.2	15	33	2.2	2.29	2.3
9	LAD prox	−0.71	0.93	0.79	1.65	1.71	19	36	1.9	2.58	2.86
10	LAD med	−5.58	0.93	0.89	1.33	2.78	10	29	2.9	3.08	3.21
11	RCA med	1.05	0.98	0.9	3.01	2.33	36	86	2.4	2.44	2.5
12 *	LAD dist	−4.51	0.89	0.72	1.88	2.34	22	45	2.1	2.84	2.49
13	LAD dist (post stent)	−3.93	0.93	0.85	1.72	2.46	22	55	2.5	2.89	2.93
14	LAD prox	−3.65	0.79	0.53	1.74	1.9	38	46	1.2	2.67	1.7
15	LCx dist (OM)	2.6	0.62	0.54	1.09	1.08	33	36	1.1	1.27	1.27
16	LCx dist (OM)	1.84	0.98	0.91	3.46	2.27	26	62	2.4	2.68	2.81
17	LAD prox	−5	0.89	0.79	1.74	2.31	15	31	2.1	2.64	2.37
18 *	LAD med	−6	0.74	0.62	1.21	1.3	31	43	1.4	1.59	1.7
19	LAD med (post stent)	−6	0.87	0.76	1.69	2.74	24	57	2.4	3.37	2.92

* Cases No. 8, No. 13 and No. 19 are the same vessels as No. 7, No. 12, and No. 18 after stent implantation. ** Hydrostatic pressure difference is the difference between the hydrostatic pressure between the pressure sensor and the tip of the catheter at the coronary orifice (hydrostatic pressure offset). *** Corrected values are calculated with the corrected hydrostatic pressures. RCA: right coronary artery; LAD: left anterior descending coronary artery; LCX: left circumflex artery; OM: obtuse marginal branch; prox: proximal vessel segment; med: medial vessel segment; dist: distal vessel segment; Pd/Pa: distal coronary pressure at rest/aortic pressure at rest; FFR: fractional flow reserve; CFR_p-3D_: coronary flow reserve calculated from intracoronary pressure data and 3D anatomical parameters; APV-B: average peak velocity at rest, APV-P average peak velocity during vasodilatation measured by Doppler wire; CFR_Doppler_: coronary flow reserve measured by Combowire; RRR_p-3D_: resistive reserve ratio calculated from intracoronary pressure data and 3D anatomical parameters; RRR_Doppler_: resistive reserve ratio measured by Combowire.

## Data Availability

The data that supports the findings of this study are available at the authors.

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
