# Peer review of "Pressure- and 3D-Derived Coronary Flow Reserve with Hydrostatic Pressure Correction: Comparison with Intracoronary Doppler Measurements"

_jpm, 2022, doi:10.3390/jpm12050780_

Round 1
Reviewer 1 Report
In the present manuscript, Tar et al aimed to an angiography-derived method for CFR and MRR calculation. They included 19 vessels. The 3d Reconstruction and fluid dynamic equations were used for the calculation of flow at rest and during hyperemia and therefore to derive the CFRp-3D. A combo wire was used for the invasive measurement of flow velocity, Pd, Pa and FFR. The authors corrected the pressure-derived indices for the hydrostatic pressure showing a good correlation between the angiography derived and the invasively measured indices.
Here are my comments:
#1
The MRR is a novel index, derived from continuous thermodilution and validated against Doppler. The formula used for Doppler Derived MRR in the manuscript from De Bruyne et al (JACC, 2021) is the following
MRR= (CFR/FFR)*(Pa,rest/Pa,hyp
Therefore, the authors should use the same formula when referring to the Doppler MRR.
However, although MRR can be calculated with other tools, its main clinical application remains related to continuous intracoronary thermodilution, as the method allows a direct volumetric quantification of absolute resting and hyperemic flow.
I would strongly suggest the author to remove the MRR calculation as objective of the manuscript and rather to focus on:
- The calculation of CFR
- The calculation of Resistance Reserve Ratio, by considering the ratio between the baseline microvascular resistance calculated by doppler (bMR= Pd,rest/APVrest) and hyperemic microvascular resistance (hMR= Pd, hyperemia/APVhyperemia) since the study is entirely focused on intracoronary doppler.
#2
In the methods paragraph it is necessary to add a section concerning the calculation of the Doppler-derived indices.
# 3
Please add the reference (PMID 33528358) as it shows the correlation between continuous thermodilution and doppler-derived CFR.
#4
Page 2. In its general form, MRR equals RRR divided by the resting pressure ratio between the distal and proximal coronary pressure (Pd, rest /Pa, rest).
This is wrong. In its general form MRR equals CFR divided by the ratio of Pd hyperemia and Pa rest. Eventually when Pa rest is equal to Pa hyperemia MRR equals CFR divided by FFR.
#4
Page 4. Formula (2) is incorrect.
#6
Page 9. Please correct “week” with “weak”.
#7
The website cannot be reached
#8
Given the small sample size the word “validation” should be removed from the title. A validation of such a tool requires a much higher number of patients.
Author Response
Responses to the Reviewer 1:
We are grateful to the Reviewer for the constructive criticism and thorough revision.
#1
The MRR is a novel index, derived from continuous thermodilution and validated against Doppler. The formula used for Doppler Derived MRR in the manuscript from De Bruyne et al (JACC, 2021) is the following
MRR= (CFR/FFR)*(Pa,rest/Pa,hyp)
Therefore, the authors should use the same formula when referring to the Doppler MRR.
However, although MRR can be calculated with other tools, its main clinical application remains related to continuous intracoronary thermodilution, as the method allows a direct volumetric quantification of absolute resting and hyperemic flow.
I would strongly suggest the author to remove the MRR calculation as objective of the manuscript and rather to focus on:
- The calculation of CFR
- The calculation of Resistance Reserve Ratio, by considering the ratio between the baseline microvascular resistance calculated by doppler (bMR= Pd,rest/APVrest) and hyperemic microvascular resistance (hMR= Pd, hyperemia/APVhyperemia) since the study is entirely focused on intracoronary doppler.
We agree that the novel MRR index is mainly related to continuous intracoronary thermodilution in clinical research, therefore, according to the Reviewer’s suggestion, we removed the MRR calculation as an objective of the manuscript and we rather focused on the calculation of CFR and RRR.
In the introduction, we just mention the MRR as a further possibility for the evaluation of microvascular function with the suggested formula (MRR= (CFR/FFR)*(Pa,rest/Pa,hyp) from the publication of De Bruyne et al. (JACC, 2021). On the basis of the 3rd instruction of the Reviewer, we added a new reference to this section: PMID 33528358 (Gallinoro E, Candreva A, Colaiori I, Kodeboina M, Fournier S, Nelis O, Di Gioia G, Sonck J, van 't Veer M, Pijls NHJ, Collet C, De Bruyne B. Thermodilution-derived volumetric resting coronary blood flow measurement in humans. EuroIntervention. 2021 1;17(8):e672-e679.).
#2
In the methods paragraph it is necessary to add a section concerning the calculation of the Doppler-derived indices.
Thank you for the useful advice. We added the proposed section about the calculation of the Doppler-derived indices to the methods paragraph:
“2.4 Calculation of the Doppler-derived indices
The resistance of the microvasculature (MR) was defined as the ratio of the distal coronary pressure divided by the distal coronary flow rate both in the basal state and during hyperemia:
where bMR: basal microvascular resistance, hMR: hyperemic microvascular resistance and APV-B: basal average peak velocity, APV-P: peak average velocity measured by the ComboWire during basal and hyperemic flow (see on Figure2)
The resistive reserve ratio (RRR) as the index of the ratio between basal and hyperemic microcirculation resistance [13, 14] was calculated as follows:
(3)”
# 3
Please add the reference (PMID 33528358) as it shows the correlation between continuous thermodilution and doppler-derived CFR.
As we mentioned at question #1, we added the reference PMID 33528358 (Gallinoro E, Candreva A, Colaiori I, Kodeboina M, Fournier S, Nelis O, Di Gioia G, Sonck J, van 't Veer M, Pijls NHJ, Collet C, De Bruyne B. Thermodilution-derived volumetric resting coronary blood flow measurement in humans. EuroIntervention. 2021 1;17(8):e672-e679.) to the appropriate section.
#4
Page 2. In its general form, MRR equals RRR divided by the resting pressure ratio between the distal and proximal coronary pressure (Pd, rest /Pa, rest).
This is wrong. In its general form MRR equals CFR divided by the ratio of Pd hyperemia and Pa rest. Eventually when Pa rest is equal to Pa hyperemia MRR equals CFR divided by FFR.
We strongly agree that MRR equals CFR divided by the ratio of Pd hyperemia and Pa rest, however it is also true that MRR equals RRR divided by the resting pressure ratio between the distal and proximal coronary artery pressure (Pd, rest /Pa, rest), as this is the rearranged version of the (omitted) formula (2), see below.
#4
Page 4. Formula (2) is incorrect.
The formula (2) quoted the Equation (3b) of the Appendix of the mentioned De Bruyne et al (JACC, 2021) publication:
MRR = RRR * (Pa, resting / Pd, resting), (2)
vs
MRR = RRR . (Pa,rest / Pd, rest) Equation (3b)
However, we have omitted this formula, as we focused on the RRR calculations instead of the MRR according to the Reviewer’s suggestion.
#6
Page 9. Please correct “week” with “weak”.
Thank you, we have corrected this typo on page 9.
#7
The website cannot be reached
The site https://coronart.unideb.hu is apparently working with our current browser settings. Previously, the server may have been temporarily down.
#8
Given the small sample size the word “validation” should be removed from the title. A validation of such a tool requires a much higher number of patients.
We kindly accept the opinion of the Reviewer regarding the meaning of “validation”, therefore we change the title to:
“Pressure- and 3D-Derived Coronary Flow Reserve with Hydro-static Pressure Correction: Comparison with Intracoronary Doppler Measurements”
Reviewer 2 Report
This is a well written, interesting and somehow complex to read pilot study, where author present the validation of a pressure and 3D-derived coronary flow reserve (CFR) corrected to the hydrostatic pressure. The proposed correction is interesting and once applied the correlations are much stronger. The main limitation of the study is the number of arteries included and the population where this methodology could be used (patients with single coronary lesions), but the authors already acknowledge these limitations.
Minimal comments/suggestions are made:
Authors present the results of cases with good quality hyperemic and resting pressure and Doppler traces. Which is the feasibility of this methodology? Could this be a limitation for the application of this methodology?
Figure 1. On description 3D anatomical parameters are supposed to be on the bottom left panel.
Figure 2. There is no description of “B” and no abbreviations are presented on description.
Author Response
Responses to the Reviewer 2:
We thank very much for the review.
This is a well written, interesting and somehow complex to read pilot study, where author present the validation of a pressure and 3D-derived coronary flow reserve (CFR) corrected to the hydrostatic pressure. The proposed correction is interesting and once applied the correlations are much stronger. The main limitation of the study is the number of arteries included and the population where this methodology could be used (patients with single coronary lesions), but the authors already acknowledge these limitations.
Minimal comments/suggestions are made:
Authors present the results of cases with good quality hyperemic and resting pressure and Doppler traces. Which is the feasibility of this methodology? Could this be a limitation for the application of this methodology?
Despite the thorough execution of the invasive Doppler investigation, good quality Doppler traces can only be achieved in 80-90% of the cases. This definitely can be a limitation of the use of intracoronary Doppler measurements.
In this study, we applied the Doppler measurements as the reference of our new calculation method based on intracoronary pressure measurement. According to our experience, precise pressure traces can be achieved more frequently than appropriate Doppler traces.
Figure 1. On description 3D anatomical parameters are supposed to be on the bottom left panel.
We wanted Figure 1 to be didactic by placing the formula of the flow calculation to the middle of the bottom panel, just under the corresponding scheme about CFR calculation.
Figure 2. There is no description of “B” and no abbreviations are presented on description.
Thank you very much for the useful remark. We added the explanation of “B” to the legend: “basal” flow vs. peak flow “P” (at maximal hyperemia).
Round 2
Reviewer 1 Report
Please remove the keyword intravascular lithotripsy
Author Response
Response to the reviewer:
Thank you very much for the review. We have contacted a native English speaker colleague and made additional changes in the manuscript regarding his suggestions.